# Effects of Lifting Method, Safety Shoe Type, and Lifting Frequency on Maximum Acceptable Weight of Lift, Physiological Responses, and Safety Shoes Discomfort Rating

**DOI:** 10.3390/ijerph17093012

**Published:** 2020-04-26

**Authors:** Fares F. Alferdaws, Mohamed Z. Ramadan

**Affiliations:** Industrial Engineering Department, Faculty of Engineering, King Saud University, P.O. Box 800, Riyadh 11421, Saudi Arabia

**Keywords:** safety shoes, manual material handling (MMH), work related musculoskeletal disorder (WRMD), lifting frequency, lifting method, heart rate, respiration responses

## Abstract

This study aimed to investigate the physical effects of precision lifting tasks on the maximal acceptable weight of a lift (i.e., psychophysiological lifting capacity where the workers adjust the lifting weight in order to work without any fatigue or strain at the end of the work while wearing common safety shoe types). Additionally, the physical difference between the precise and non-precise lifting conditions associated with wearing safety shoes were assessed by respiration responses and shoe discomfort ratings. To achieve the objective of the study, ten healthy male workers were selected by age (between 25 to 35 years old). Their anthropometric characteristics, including knuckle height, knee height, and body mass index (BMI), were measured. A three-way repeated measures design with three independent variables was used; the variables included—the (1) lifting method (precise and non-precise), (2) lifting frequency (1 and 4 lifts per min), and (3) safety shoe type (light-duty, medium-duty, and heavy-duty). The physiological response variables and one of the subjective factors of this study were—(1) respiration responses, and (2) shoe discomfort rating, respectively. The data were analyzed using the Mauchly’s test of sphericity, Shapiro–Wilk normality test, and analysis of variance (ANOVA). The results showed that the use of heavy-duty safety shoes typically increased the shoe discomfort rating under precise lifting methods. Additionally, the lifting frequency was determined to be one of the main factors affecting respiratory responses and shoe discomfort rating. This study also found that respiration responses rose on four lifts per min as compared to 1 lift per min, regardless of the lifting method type. This study indicated that the replacement of some types of ordinary safety shoes used in some workplaces with those selected appropriately might significantly reduce the rating effort required to lift objects or tools. However, the benefits should be carefully evaluated before replacing the safety shoes.

## 1. Introduction

Manual material handling (MMH) is one of the major causes of severe industrial injury [1]. MMH needs to be lifted but usually includes lifting, carrying, pushing, pulling, climbing, and rotating, all of which pose a risk of back injuries [2]. Working conditions, such as walking and standing on a hard surface, can increase the development of musculoskeletal disorders. The shoe types that are worn and the amount of work performed can affect muscle movement, which is critical in a professional setting [3], where continuous carrying might cause muscle fatigue in the upper limbs [4]. At the interface between the floor and the musculoskeletal system, safety shoes might play an essential role in the well-being of employees [5]. As modern work often requires rapid physical exertion and the need for psychological treatment (source of psychological stress) [6], in the past few decades, the problems caused by frequent lifting and lowering tasks have increased rapidly [7]. Any work involving heavy manual material handling might involve a high risk of injury at work.

In 2017, there were 1.1 million non-fatal occupational injuries and illnesses, of which injured or sick workers had at least one day of work and rest. This was roughly the same as in 2016. The incidence rate in 2017 was 98.0 per 10,000 full-time equivalent workers [8]. The exchange rate for 2016 was 100.4. In 2017, the median from work to rest was 9 days, the same as that in 2016. Transportation and material moving workers had the highest number of cases with days away from work in 2017 (195,800 cases). The incidence rate for these workers fell to 235.3 cases per 10,000 full-time equivalent workers in 2017, down from 244.0 cases in 2016 [8]. As for the construction industry, MMH tasks are widespread on construction sites and are one of the main causes of musculoskeletal diseases in workers [9], MMH’s work accounts for a large proportion of the 1.1 million cases of musculoskeletal diseases reported annually in the United States. Work related musculoskeletal disorders (WMSD) usually involve strains and sprains in the lower back, shoulders, and upper limbs [10]. Calzavara et al. [11] indicates that people recognize that MMH in warehouses expose workers to high risks of musculoskeletal disorders.

In addition, when employees remain under unhealthy work conditions for a long time, accumulated metabolic waste can cause pain, and excessive and repeated operations can cause workers to fatigue and develop morbid conditions. Unhealthy work environments and manual procedures can lead to WMSD—the most common job-related occupational injuries. WMSD is a disease or condition of muscles, nerves, tendons, joints, cartilage, and intervertebral discs, such as low back pain, neck, and shoulder syndrome, and sprains [12,13,14]. WMSD is the leading cause of nonfatal occupational injuries and might result in temporary or permanent disability. This pressing issue about WMSD is becoming more and more prominent. For example, 25% of workers in Europe suffer from back pain and 23% endure muscle pain [15].

The risk of WMSD raises when an object is heavy, tricky, or massive, and is growing from the ground level, which requires repetitive lifting, or when the task demand exceeds the physiological capacity of the person [16,17]. Correspondingly, increased lower back pains is spotted in workers through heavy lifting loads [18]. Nevertheless, the occurrence of lower back injuries can be decreased by 30% if the worker’s strength or physical capacity is similar to the particular occupational requirements of the risk [19]. To evaluate the assessment of physical task demand and to reduce the high incidence of lower back injuries, MMH guidelines, and formulae have been developed [20]. Furthermore, to minimize the injuries without back pain in lifting tasks, the National Institute for Occupational Safety and Health (NIOSH) developed an equation to assist in evaluating lifting demands, called the NIOSH equation. However, this formulation is not adequate and accurate to take into consideration, due to the full range of alterations in lifting that takes place professionally every day.

The effect of whole-body restriction on manual tasks has been found to significantly reduce physical capacity [21,22]. Moreover, the lifting task frequently needs an accurate object placing to obtain an effective task execution and restrict damage to the object and its direct circumference. Precision, in terms of restrictions on object placement, has been found to significantly decrease physical demand and lifting task durations [23,24]. Nonetheless, few prior studies on the impact of precision on lifting capability have been conducted. It has generally been found that as lift height increases, the maximum acceptable weight of the load tends to decrease. In addition, the lifted weight is also a function of the lift rate; as the lifting frequency (lifts/min) increases, the maximum acceptable weight decreases [25]. Moreover, precision, in terms of limb position of force during isometric single-joint contractions, has been found to significantly reduce endurance capacity. In addition, the precision lifting and carrying while wearing safety shoes was not reported, but has been found to significantly reduce lifting capabilities while an individual performs manual handling tasks. Therefore, this study investigated the physical effects on the maximal acceptable weight of a lift, while wearing ordinary safety shoes.

Previous studies were broadly applied to specify extreme lift ability and maximum acceptable weight of lift (MAWL) for a worker, which a worker can lift it to determine safe lifting [26]. Previous studies have concluded that estimating the dynamic strength is more significant than estimating static strength in maximal lift capacity [25,26,27]. Furthermore, the MAWL for lifting is a superior predictor of maximal lift capacity, when compared to other strength assessments, due to the identical task characteristics, including the lifting technique of the object [19,28,29,30,31,32]. Similarly, repetitive lift endurance performance was shown to be effective in predicting the risk of musculoskeletal disorders and therefore is a strong predictor of physical capacity [28,29,33,34,35,36,37,38].

In many industrial environments, there might be a variety of different and often complex MMH tasks. The maximum acceptable weight can be determined directly by the psychophysical method. This method involves designing tasks that are suitable for most workers, and in which individual controls are given to adjust the weight, force, or frequency of a particular task to the maximum acceptable value they can perform on them [39,40].

Additionally, a psychophysical study was implemented by Snook and Ciriello [41] to investigate the effects of lifting boxes with and without handles. Wu [42] studied the effect of asymmetric lifts on MAWL, and found that MAWL decreased as the asymmetry angle increased. However, heart rate, oxygen uptake, and perceived exertion rating (PER) did not change. Moreover, MAWL decreased significantly with increasing lifting frequency and the physiological costs (heart rate and oxygen uptake). Additional study examined by Lee [43] found that the MAWL decreased with increasing frequency and container width or length dimension. A psychophysical study by Maiti and Ray [44] was carried out on ten adult Indian female construction workers and eight domestic workers. Both groups psychologically rated this work as a moderate to severe category. Effects of body mass index (BMI), box size, lift frequency, and vertical distance on MAWL of Indian male workers were studied [45]. The results showed that MAWL decreased significantly with increasing frequency, box size, and vertical distance, and increased with the increase in worker BMI.

The physiological approach concerns the whole body fatigue, such as oxygen consumption, heart rate, and rate of energy expenditure. This method focuses on designing tasks such that the physiological response of the body would be within acceptable limits [46,47]. Oxygen consumption, heart rate, and rate of energy expenditure are the major physiological measurements used to identify the maximum work performed without fatigue. Energy cost, in the case of lifting or any other kind of MMH tasks, was predicted by the physiological models that can be continuously performed without accumulating excessive physical fatigue. These works have been done by many researchers [48,49]. Some other researchers [9,50] partitioned the total energy expenditure into smaller components, which, when added, gave the required values. Asfour [51] developed prediction models to predict energy expenditure for lifting or lowering on the sagittal plane. Other works reported by Simpson et al. [52] determined how changes in load quality affect a woman’s heart rate (HR), posture, and subjective response during long walks. The results showed that HR was not significantly affected by load mass or walking distance. Increased load mass and distance substantially affected upper body posture, RPE, and discomfort ratings.

Geraldo et al. [53] demonstrated that Cheyne-Stokes Respiration (CSR) is a periodic respiration associated with periodic oscillations in blood pressure (BP) and heart rate (HR). In their study, the effects of lifting variables such as lifting load, lifting frequency, vertical height, horizontal distance, laboratory conditions and box size on oxygen intake and heart rate were considered. The results showed that lifting load, increasing frequency, vertical height, horizontal distance, laboratory environment, and load size had a significant impact on oxygen intake. In order to assess the capabilities of workers, it was also necessary to accurately simulate MMH tasks. This experiment aimed to investigate the effects of manual lifting methods and lift frequency while wearing safety shoes, on lifting capacity, physiological responses, i.e., respiration responses, heart rate, and the reported level of foot stress (shoe discomfort rating).

## 2. Methods

### 2.1. Study Design

Previous research work have indicated strong relationships among physiological responses, MAWL, and workplace criteria in terms of lifting frequencies [25,42,43,44,45]. However, only a few studies have tried to investigate them by considering wearing safety shoe types [38,54,55]. No research has been reported on their effects while lifting a load in a precision manner. Therefore, an experimental study design was used for assessing the changes in physiological responses and discomfort rating of individuals associated with MAWL, while wearing different types of safety shoes under different lifting frequencies. Thus, the study design was similar to those used in previous research work [38,54] that provide easily comprehensible statistical representations of results.

### 2.2. Participants

The sample size was determined using the effect size (partial eta-squared “η^2^”) for an important dependent measure MAWL in a similar study [54]. Using the found η^2^ of 0.25, and Type I error of 0.05 for 80% of power, the correlation among repeated measures of 0.5 and nonsphericity correction of 1, the study required a sample size of at least ten participants in each level, which was the number of recruited participants in the experiment, as determined by the G*Power software (version 3.1.9.7, Heinrich-Heine-Universität, Düsseldorf, Germany) [56].

Ten male workers from the university with a mean age (standard deviation, SD) of 29.70 (3.34) years were hired from the working population. The participants’ mean weight (SD) was 72.20 (7.21) kg, mean height was 167.30 (7.13) cm, mean BMI was 25.79 (2.03) kg/m^2^, mean knee height was 46.73 (1.32) cm, and mean knuckle height was 68.40 (1.32) cm. None of the participants had experienced any back or lower and upper limb disorders, heart disease symptoms, or breathing complications. All trials were conducted in January and February, 2019, and participants’ time was compensated with 50 Saudi Riyals/trial. All participants were informed of the purpose and experimental steps of the experiment and signed a consent form approved by the Associate Dean of the Graduate School of Research and Scientific Research at King Saud University (Scientific Research Director—Ethics Committee under ethical code # E-18-3752). This process was completed by submitting a written proposal with all research details, including scope, goals and objectives, methods, protocols, participants’ details, requirements, and agreements. In addition, workers’ written consent was obtained after explaining the study goals and objectives and their roles and responsibilities as participants (in their native language, mainly Urdu and Hindi). The principle investigator also communicated the rights of participants to withdraw from the study at any time.

### 2.3. Measurement and Instrumentation

#### 2.3.1. Anthropogenic Measurements

A balance scale (Seca 708, 0.1–200 0.1 kg) was used to measure weight (kg) while wearing lightweight clothing without any footwear. Height (cm) was measured by ensuring that the participant was in an upright position in front of the front wall and the heels were in contact with each other. The body mass index (BMI) was calculated using the formula: BMI = mass (kg)/height (m)^2^. The Siber Hegner anthropological instrument was used in this experiment. The instrument included the following—(1) fixed anthropometry (0–2100 mm with straight probe and curved measurement branches); and (2) fiberglass belts (Dean, 0–1500 mm). When measuring these dimensions, the procedures used followed these references [57,58].

#### 2.3.2. Maximum Acceptable Weight of Lift (MAWL)

The maximum acceptable weight of lift (MAWL) was determined psychologically. It was the maximum weight that the participant repeatedly increased or decreased for a given lifting condition, according to the participant’s perception and when not overworked or fatigued [59,60,61,62]. Maximal lifting capacities were specified prior to the maximum acceptable lifting weight quantity, relative to the starting lift loads [34]. Two starting lift loads were pre-set for the participant, a light box load (~33% of their maximal lifting capacity) and a heavy load (~95% of their maximal lifting capacity). Participants were randomly assigned to start with the light or heavy box and then instructed to make as many weight adjustments for increments or decrements in mass, to determine the maximal acceptable weight of lifting for a period of fifteen minutes. Then, a participant continued working for another 5 min for the measurements to be considered for data analysis. At the end of each session, a minimum of 5-min rest was allowed between each lifting trial to ensure recovery. The participants were blinded to the mass adjustments and the final maximal acceptable weight of lift.

#### 2.3.3. Respiration Responses

A Moxus Modular Metabolic cart using legendary metabolic CD-3A and S-3A Gas Analyzers (AEI Technologies, Inc., Bastrop, TX, USA) was used to measure oxygen uptake and respiration responses. The main elements of the Moxus Metabolic unit included a carbon dioxide analyzer (CD-3A), an oxygen analyzer (S-3A/I), a carbon dioxide analyzer (CD-3A), a 4.2-L active mixing chamber, a flow control pump (R-1), a canopy Pump, a canopy Hood, calibration and reference gases, a face mask worn by the participants (a leakage tested), and a laptop with an interface software (to calibrate and collect data for further analysis). The mask was attached to the central unit through two flexible tubes. A computer with software to record and analyze data respiration responses and heart were associated with the central unit. Finally, the Moxus cart measured lung ventilation through a turbine volume sensor mounted on the air inlet.

While the participants were lifting and carrying the MAWL and walking on a treadmill (OAC297-OLYMPIA; Olympia Fitness Systems, Gujarat, India) [63], the measurements of their heart rates, oxygen uptakes, and respiration responses were measured and recorded every 10 s. The instrument was calibrated according to the manufacturer’s instructions. Averages of the last five-mins of the respiration responses were computed and considered for statistical analysis in this study.
Respiration Rates (RR (breaths/min)): The respiration rate was the number of breaths taken in 1 min, measured as the number of times the participants’ chest rose in one min.Minute Ventilation (Ve (L/min)): It was measured as the volume of air a person can exhale in liters during a breathing process, in one min.CO_2_ production (V_CO_2__ (mL/min)): It was measured as the volume of carbon dioxide that a person breathed out after transporting oxygen through the body.Oxygen Uptake/body mass (V_O_2__/kg): It was measured as the amount of oxygen consumption a person can utilize during the breathing process in one min, per person’s body mass.

#### 2.3.4. Heart Rates (beats/min)

Many researchers have shown that heart rate is directly proportional to the workload and lift frequency [59,60,61]. The heart rates of the participants were measured and monitored using a Polar monitor (Polar Kemple Co, Finland). It was covered tightly with a conductive gel and placed on the participant’s sternum, which exposed to the skin clean.

#### 2.3.5. Safety Shoes Discomfort Rating (SSDR)

SSDR was defined as the level of discomfort while wearing safety shoes during lifting activity. This scale could be a good comparison of different types of safety shoes [64,65]. Participants in the experiment were asked to verbally report their levels of discomfort when wearing safety shoes, at the end of each session, where (1) was the least discomfort, and (5) was the highest discomfort. This measure reflected the workers’ sense of discomfort while wearing safety shoes. The scale was translated to the worker’s language and was posted on the chamber wall in front of the participant, during the experimental session.

### 2.4. Experimental Variables

The independent variables in this study were: (1) lifting method (precise and non-precise), (2) two levels of lifting frequencies (1 lift per min and 4 lifts per min), and (3) three types of safety shoes (light safety shoes, medium safety shoes, and heavy safety shoes).

#### 2.4.1. Lifting Method

Two different lifting methods (precise and non-precise) were used in this study. Precise and non-precise lifting conditions differed only when the box was placed on the shelf, precisely by placing the box on the limited wooden frame, as shown in Figure 1a. In a non-precise condition, the wooden frame was not used, see Figure 1b.

#### 2.4.2. Lifting Frequency

The frequencies employed in this study were 1 and 4 lifts/min. In addition, many studies have investigated different lifting frequencies of 1, 4, and 5 lifts per min (e.g., [38,45,51,55,60]); which allows for a comparison.

#### 2.4.3. Safety Shoe Types

Three safety shoe types were used in this study (manufactured by Shelterall Co., Italy). These three types complied with the Saudi Standard Specification No. SASO/ ISO 20345 /2007. They were lightweight, medium, and heavy-duty safety shoes. Lightweight safety shoes were similar to ordinary leather shoes and were considered a reference (it included a full leather double seal, a padded collar, a rubber sole, a steel toe cap, and was low weight—0.9 kg). Additionally, medium safety shoe types were also used. These were made of full leather, had a double-density padded collar, a polyurethane molded sole, a low-top steel toe cap, and weighed 1.05 kg. Finally, a heavy safety shoe type was used. These were made of wax full grain leather, had a double density padded collar, a polyurethane molded sole, a high cut double steel toe, and weighed 1.45 kg. These safety features are shown in Table 1. Moreover, Figure 2 shows the three types of safety shoes that were used. The main researcher instructed to the participants to wear the most suitable pair of safety shoes, all of which could be used in the laboratory in various sizes. Mills et al. [64] used a ranking scale to assess the feeling of discomfort, where the participants were asked to rank their discomfort.

### 2.5. Experimental Design

In this study, in order to determine the effects of safety shoe type, lifting method and lifting frequency of respiration responses, and perceived exertion rating were measured. A three-way repeated measures analysis of variance (ANOVA) design with three independent variables, two physiological responses and one subjective factor were utilized. Therefore, the study had 12 experimental conditions corresponding to a combination of 12 independent variable levels. The test sequence was randomly assigned. After confirming the normality of the data using the Shapiro-Wilk parameter test and the Mauchly’s Test of Sphericity, ANOVA statistical analyses were performed using the SPSS software version 22.0 (IBM Corp., Armonk, NY, USA), taking into account the significant differences between the data (*p* < 0.05). Tukey’s test was used to distinguish and discover significant differences among shoe-type factor levels.

### 2.6. Experimental Procedures

After obtaining the consent of each participant, the participant’s anthropometric data and maximum amount of weight were measured and recorded. Then, a participatory plan was established. In order to familiarize participants with devices and factors, protocols and procedures were developed and a brief demonstration of how to properly lift the box and adjust the weight was provided. The trials were started in the order of planning in a random fashion. On the first day, each participant was trained, familiarized with the experimental protocol, and then proceeded to the manual lift. All participants were exposed to all treatments. Moreover, participants were randomly and sequentially required to wear the pair of safety shoes specified in the experiment. All participants performed the lifting tasks at two different frequencies of lift (1 and 4 lifts/min), wearing one of the three types of safety shoes (light, medium, or heavy safety shoes) with two different lifting methods (precise and non-precise). Each participant was asked to refrain from smoking or eating for at least two hours prior to each data collection session. They were also asked to avoid participation in any strenuous physical activity prior to the experimental sessions, and to get their normal amount of sleep. Participant clothing was controlled by instructing them to wear their own, light, working clothes.

At the beginning of each data collection session and prior to starting the experiment, a Mous’s mask was worn on the head to measure respiratory responses. The participant started walking on the treadmill and lifting the wooden box under the supervision of the experimenter and helper, who lowered the wooden box from the participant’s shoulder to his knuckle and adjusted the weight lifted. The test was terminated if the participant showed any signs of musculoskeletal disorder, or if the participant requested the experimenter to stop the session. During the weight adjustment process, each participant asked the helper to increase or decrease the weight, until reaching his selected weight (maximum acceptable weight) in both treatments, of different starting assigned weight.

After the 5-min pre-work period, the participant started to lift the wooden box. Each participant was allowed 15 min to determine the maximum acceptable weight of lift when lifting at frequencies of 1 or 4 lifts/min under one of both lifting methods (precise and non-precise), while wearing one of the different shoe types. Participants were encouraged to make weight adjustments by starting with a random choice of box weights.

Finally, each participant was asked to continue lifting using the maximum acceptable weight for another 5-min period, without making any further adjustments using the entire session, all physiological responses were measured and monitored. If heart rates reached the maximum recommended limit before the session was completed, the experiment was stopped (maximum heart rate = 220—participant’s age). A second round was completed after a five-min recovery period. At the end of each session, the participant was asked to remain in the laboratory environment, under observation, for an additional five mins, during which his recovery heart rate was observed and recorded. After each trial, the participant verbally ranked his shoe discomfort feeling.

## 3. Results

Three-way repeated measures design was carried out to test the effect of the lifting method and lifting frequency while wearing different safety shoes, on the dependent variables.

### 3.1. MAWL

ANOVA results showed that the three independent variables—lifting method (LM), F (1,9) = 189.357, *p* < 0.000, η^2^ = 0.955; lifting frequency (LF), F (1,9) = 284.832, *p* < 0.000, η^2^ = 0.969; and shoe type (ST) F (2,18) = 4.170, *p* < 0.032, where η^2^ = 0.317, as shown in Table 2. First, it was shown that MAWL was significantly lower at precise lifting (mean, (SD) 9.978 (0.465)), when compared to non-precise lifting (mean, (SD) 12.095 (0.543)), as shown in Figure 3a. Second, MAWL was also significantly higher at 1 lift/min (mean, (SD) 12.398 (0.531)), when compared to the MAWL lifted at 4 lifts/min (mean, (SD) 9.675 (0.479)), as shown in Figure 3b. Third, MAWL was significantly higher while participants were wearing light safety shoes (mean, (SD) 11.305 (0.541)) when compared to the MAWL while participants were wearing heavy safety shoes (mean, (SD) 10.723 (0.527)), *p* < 0.041, as shown in Figure 3c. Moreover, no significant differences were found between MAWL when participants were wearing light or medium safety shoes (mean, (SD) 11.083 (0.469)). Additionally, no significant differences were found between MAWL when participants were wearing medium or heavy safety shoes.

### 3.2. Resperation Responses

#### 3.2.1. Respiration Rates

ANOVA results showed that only one independent variable—lifting frequency (LF), F (1,9) = 43.530, *p* < 0.000, η^2^ = 0.829; had a significant effect on the respiration rate (breaths/min), as shown in Table 3. The ANOVA showed that the respiration rates (breaths/min) were significantly higher at the 4 lifts/min lifting frequency (mean, (SD) 28.77 (1.70) breaths/min), compared to the 1 lift/min lifting frequency (mean, (SD) 24.12 (1.78) breaths/min), as shown in Figure 4.

#### 3.2.2. Minute Ventilation (Ve (L/min))

ANOVA results showed that only one independent variable—lifting frequency (LF), F (1,9) = 67.861, *p* < 0.000, η^2^ = 0.883—had a significant effect on Ve (L/min), as shown in Table 4. ANOVA showed that Ve (L/min) was significantly higher at 4 lifts/min lifting frequency (mean, (SD) 28.70 (1.70)), when compared to a 1 lift/min lifting frequency (mean, (SD) 23.40 (1.30)), as shown in Figure 5.

#### 3.2.3. V_CO_2__ (mL/min)

ANOVA results showed that only one independent variable—lifting frequency (LF), F (1,9) = 60.375, *p* < 0.000, η^2^ = 0.870—had a significant effect on V_CO_2__ (mL/min), as shown in Table 5. It showed that V_CO_2__ (mL/min) was significantly higher at 4 lifts/min lifting frequency (mean, (SD) 769.50 (54.80)), when compared to the 1 lift/min lifting frequency (mean, (SD) 583.15 (37.20)), as shown in Figure 6.

#### 3.2.4. V_O_2__/kg

ANOVA results showed that only one independent variable—lifting frequency (LF), F (1,9) = 52.163, *p* < 0.000, η^2^ = 0.853—had a significant effect on V_O_2__/kg, as shown in Table 6. The results demonstrated that V_O_2__/kg was significantly higher at 4 lifts/min (mean, (SD) 13.37 (0.94)) than 1 lift/min (mean, (SD) 10.50 (0.65)), as shown in Figure 7.

### 3.3. Heart Rate

ANOVA results showed that only one independent variable—lifting frequency (LF), F (1,8) = 16.664, *p* < 0.004 where η^2^ = 0.676; had a significant effect on heart rate (beats/min), as shown in Table 7. The results illustrated that heart rate (beats/min) was significantly higher at 4 lifts/min lifting frequency (Mean (SD) = 106.58 (2.91)), than at 1 lift/min lifting frequency (Mean (SD) = 95.17 (2.37)), as shown in Figure 8.

### 3.4. Safety Shoes Discomfort Rating (SSDR)

ANOVA results showed that three independent variables—lifting method, F (1,9) = 5.898, *p* < 0.038, η^2^ = 0.396; lifting frequency, F (1,9) = 10.796, *p* < 0.009, η^2^ = 0.545; and shoe type, F (2,18) = 100.063, *p* < 0.000, η^2^ = 0.917—had significant effects on safety shoe discomfort rating (SSDR), as shown in Table 8. First, ANOVA showed that SSDR was significantly higher in the precise lifting method (mean, (SD) 2.78 (0.09)), when compared to the non-precise lifting method (mean, (SD) 2.50 (0.09)), as shown in Figure 9. Secondly, SSDR was significantly higher at 4 lifts/min (mean (SD) 2.83 (0.32)) when compared to 1 lift/min lifting frequency (mean, (SD) 2.50 (0.09)), as shown in Figure 10. Moreover, the SSDR was also significantly higher while wearing light, medium, and heavy safety shoes (mean (SD) = 1.60 (0.14), mean (SD) = 1.88 (0.18) and mean (SD) = 4.50 (0.11)), as shown in Figure 11.

## 4. Discussion

The main objective of this study was to investigate the effects of the lifting method and lifting frequency on MAWL, safety shoes discomfort rating (SSDR), and human body physiological responses, i.e., respiration rate, Ve, V_CO_2__, V_O_2__/kg, and heart rate, while wearing different safety shoes types. The hypotheses of this study stated that lifting frequency, lifting methods, and worn safety shoes had a greater influence on increasing safety shoes discomfort rating, heart rate, and respiration responses, during a muscular activity. The influence of increased respiration rates and heart rates could be explained as increased work stress and cause a decrease in lifting capability. On the other hand, the increase in safety shoes discomfort rating (SSDR) could be because of wearing a specific safety shoe type.

As the precise lifting and carrying processes were performed by lifting and placing objects in tight space, increasing accuracy requirements most likely caused a slowing of the movement near the destination of the lift. This type of lifting increased the holding time, and participants responded by accepting lighter loads. Hence, this study proved that the lifting method, safety shoe type, and lifting frequency had significant effects on the MAWL. The outcomes of the experiment showed that increasing lifting frequency from 1 lift/min to 4 lifts/mins decreased the MAWL. This finding was found corresponding to previous studies where an increased lifting frequency from 1 lift/min to 4 lifts/min caused a decrease in mean weight lifted by 21.96%, as compared to the 22.08% reported by Chen et al. [61], 19.8% reported by Ghaleb et al. [54], and 16.67% reported by Lee [43]. Mital [62] states that the reductions in MAWL could be lower because of the differences in environmental conditions and also in the population studied. Such as, the present study had cleaning workers as participants, whereas Mital [62] had lifting workers as its participants. In general, the MAWL decreased as lifting frequency increased, which escalated the intensity of the workload.

Precision lifting was found to significantly decrease the MAWL to be lifted, which was in agreement with the finding reported in Mital and Wang [22]. In addition, precision lifting caused significantly more musculoskeletal system stress more than non-precise lifting [66,67]. Generally speaking, the MAWL decreased with increasing musculoskeletal system stress, which is an index of increasing workload intensity.

The MAWL of the participants while wearing heavy-duty or medium-duty safety shoes was significantly less than the MAWL in the light-duty safety shoes with lifting activities. This finding was supported by the findings by Al-Ashaik et al. [38] who in their studies found a 6.13% reduction in MAWL associated with wearing light-duty shoe type, compared to 11.39% reduction in the MAWL associated with this study. However, on the other hand, the safety shoe type had no significant effect on the MAWL associated with lifting activities [54].

In general, respiration rates (breaths/min) increased, as the frequency of lifting increased. A comparison of the results of the present study with other studies was not possible as there are no similar studies available in the existing literature. However, change in heart rates is always found to be associated with the change in respiration rates.

In this study, Ve (liters/min) was significantly affected by lift frequency. Results showed that Ve (liters/min) increased as the frequency of lifting increased. It was observed that the mean Ve (liters/min) at a lifting frequency of (4 lifts/min) increased by 22.65%, when compared to a lifting frequency of (1 lift/min). Additionally, V_CO_2__ (mL/min) was significantly affected in this study by the frequency of lift independent variable. Increasing lifting frequency was found to significantly increase V_CO_2__ (mL/min) rating. Results showed that V_CO_2__ (mL/min) increased as the frequency of lifting increased. It was observed that the mean V_CO_2__ (mL/min) at lifting frequency (4 lifts/min) increased by 31.96%, when compared to the lifting frequency (1 lift/min). It was challenging to compare our results on independent variables and V_CO_2__ (mL/min), since there are no similar studies in the existing literature.

Oxygen uptake (V_O_2__/kg) was significantly affected by the frequency of lift. Increasing lift frequency was found to significantly increase V_O_2__/kg. Results showed that V_O_2__/kg increased as the frequency of lifting increased. It was observed that the mean V_O_2__/kg on lifting frequency (4 lifts/min) increased by 27.33%, when compared to a lifting frequency of (1 lift/min).

The effect of the lifting frequency was the only main independent variable influence on the working heart rate (beats/min). Results showed that heart rate increased as the frequency of lifting increased. It was observed that the mean heart rate (beats/min) during a lifting frequency of (4 lifts/min) increased by 11.99%, when compared to a lifting frequency of (1 lift/min). These results agreed with the results obtained by [38,54]. Previous studies showed very close agreement with this study through different values. These difference in values in the amount of deficiency between the current study and previous studies were due to differences that existed in participants and environmental conditions. The results are shown in Table 9.

The safety shoe discomfort rating in this experiment supported the results of other studies. Safety shoe discomfort rating was affected by increased intensity of workload increasing frequency of lifting. It was observed that the mean shoe discomfort rating at 4 lifts/min lifting frequency increased by 13.20%, when compared to lifting frequency (1 lift/min). Secondly, heavy safety shoes had an increased mean safety shoe discomfort rating (SSDR), when compared to light and medium safety shoes. These results accorded with the results obtained by Al-Ashaik [38] and Ghaleb et al. [54], which showed that heavy safety shoes were associated with safety shoes discomfort rating (SSDR), thus indicating an association between safety shoes discomfort rating (SSDR) and the type of safety shoes. The results are shown in Table 10 and Table 11. Previous studies showed very close agreement. It was difficult to compare our results regarding the lifting method conditions and shoe discomfort rating because there was no similar study available in the existing literature.

## 5. Limitations

Some limitations of this study should be emphasized. First, the sample size of the participants was somehow small, limiting the generality of the results to the population. However, this trend might help designers better understand the importance of studying safety shoe types in precise manual materials handling. In addition, other respiratory response and heart rate variables could be included in such studies and could add new dimensions, when exploring the higher benefits. Additionally, the authors planned to have different worker age groups. However, the Associate Dean of the Graduate School of Research and Scientific Research refused to issue approval for including older workers in the study.

## 6. Conclusions

This study conducted an investigation into the effects of safety shoes on respiration response, heart rate, and safety shoes discomfort rating, under two lifting methods (precise and non-precise) and lifting frequency (1 and 4 lifts/min). Additionally, the study investigated three shoe types used in different industrial settings, such as light, medium, and heavy duty safety shoes. Experiments have confirmed that the use of heavy-duty safety shoes typically increase the safety shoes discomfort rating under precise lifting methods. In addition, the lifting frequency is determined to be one of the main factors affecting heart rate, respiratory response, and safety shoes discomfort rating. This study also found that respiration responses and heart rate rose on 4 lifts per min, as compared to 1 lift per min lifting frequency, regardless of the lifting method type.

This study suggests that the replacement of some types of ordinary safety shoes used in some workplaces with those appropriately selected might greatly reduce the safety shoes discomfort rating required to lift objects in a fairly tight zone. However, to get the highest benefits, recommended types should be carefully evaluated before replacing safety shoes. This research might also lead ergonomists to work more diligently to improve task and activity planning, while considering safety shoe characteristics. As no other studies have looked at the accurate and careful placement of objects in limited space, ergonomists might consider implementing the results of this paper as outlined in the recommendations, when designing a similar type of lift configuration, as that studied in this paper.

## Figures and Tables

**Figure 1 ijerph-17-03012-f001:**
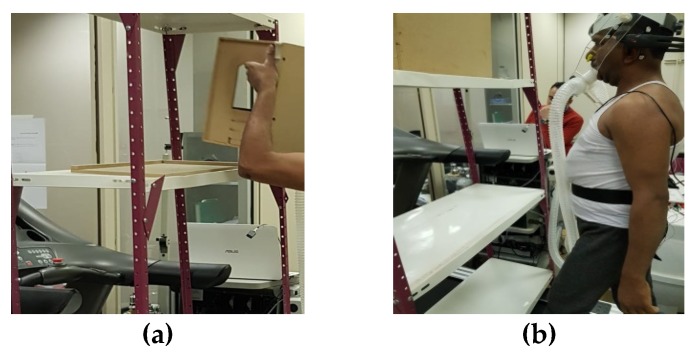
Lifting Method, (**a**) the precise lifting method; and (**b**) the non-precise lifting method.

**Figure 2 ijerph-17-03012-f002:**
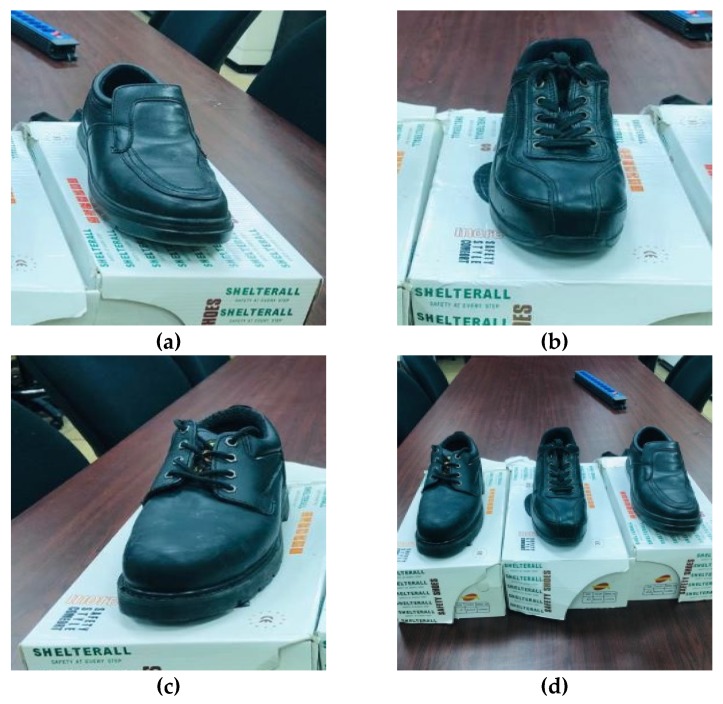
Safety shoes used in the study (**a**) light-duty safety shoes; (**b**) medium-duty safety shoes; (**c**) heavy-duty safety shoes; and (**d**) all safety shoes.

**Figure 3 ijerph-17-03012-f003:**
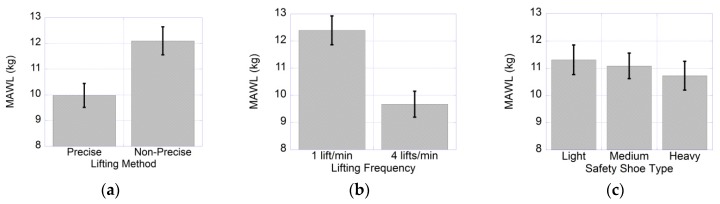
Effects of lifting method (**a**), lifting frequency (**b**), and safety shoe type (**c**) on MAWL (maximum acceptable weight of lift).

**Figure 4 ijerph-17-03012-f004:**
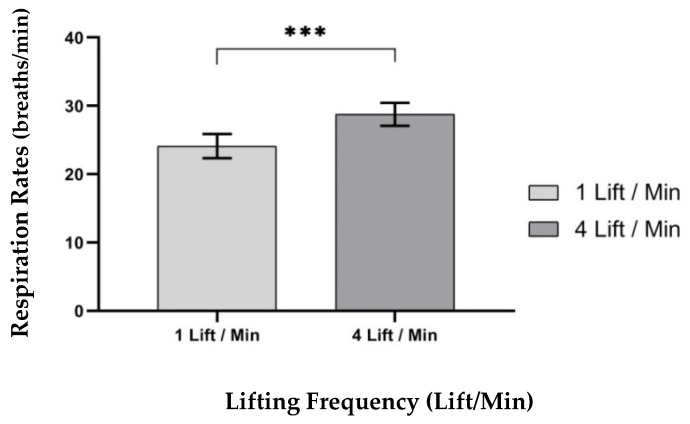
Effect of lifting frequency on respiration rates (breaths/min). Bars errors represent the standard deviations. *** *p* < 0.000.

**Figure 5 ijerph-17-03012-f005:**
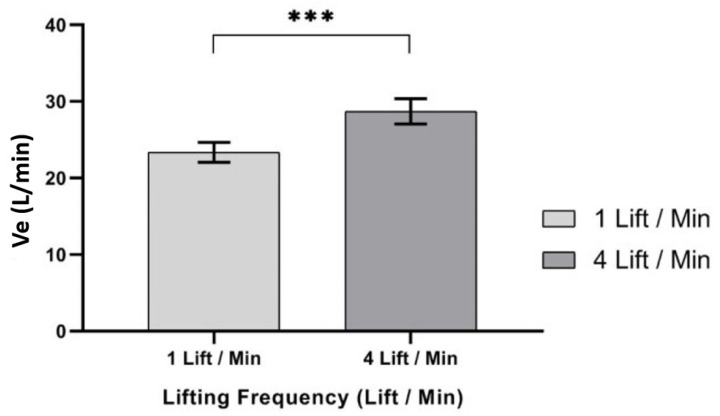
Effect of lifting frequency on Ve (L/min). Bar errors represent the standard deviations. *** *p* < 0.000.

**Figure 6 ijerph-17-03012-f006:**
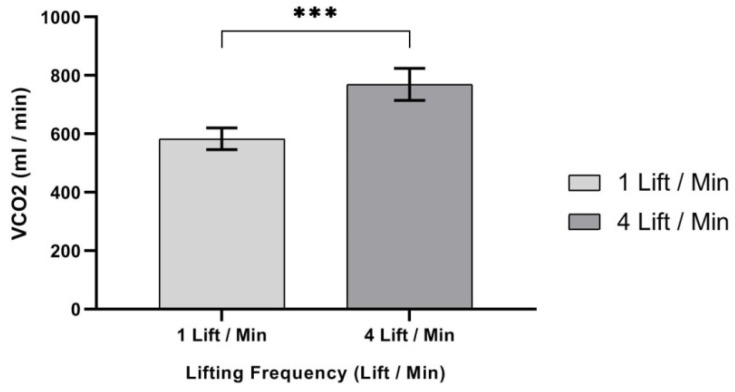
Effect of lifting frequency on V_CO_2__ (mL/min). Bar errors represent the standard deviations. *** *p* < 0.000.

**Figure 7 ijerph-17-03012-f007:**
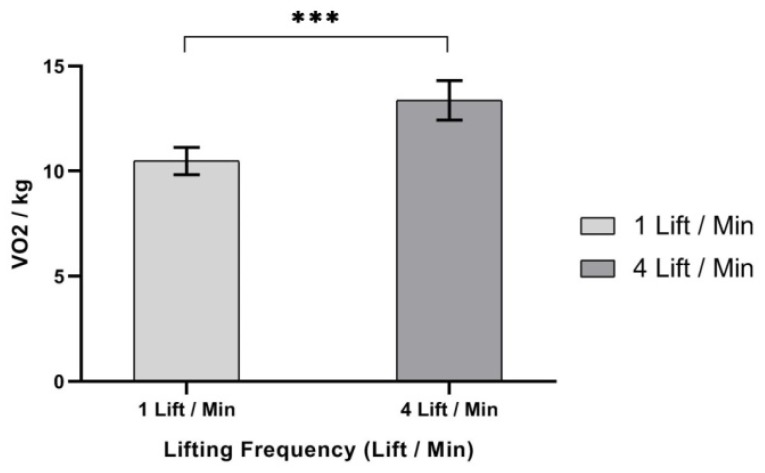
Effect of lifting frequency on V_O_2__/kg. Bar errors represent the standard deviations. *** *p* < 0.000.

**Figure 8 ijerph-17-03012-f008:**
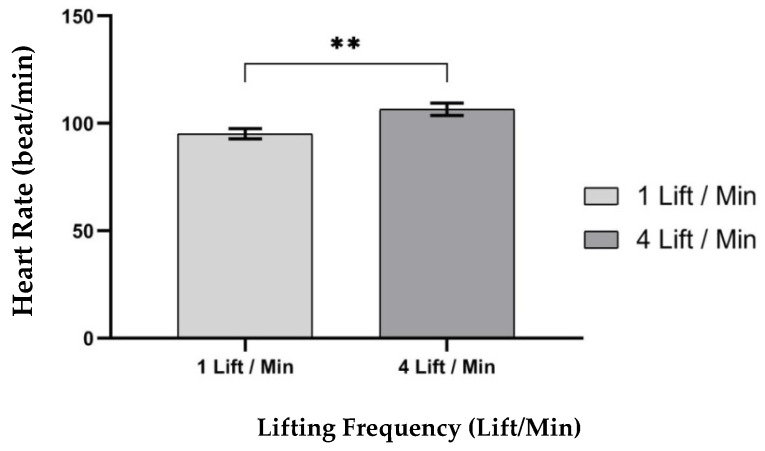
Effect of lifting frequency on heart rate. Bars errors represent the standard deviations. ** *p* < 0.005.

**Figure 9 ijerph-17-03012-f009:**
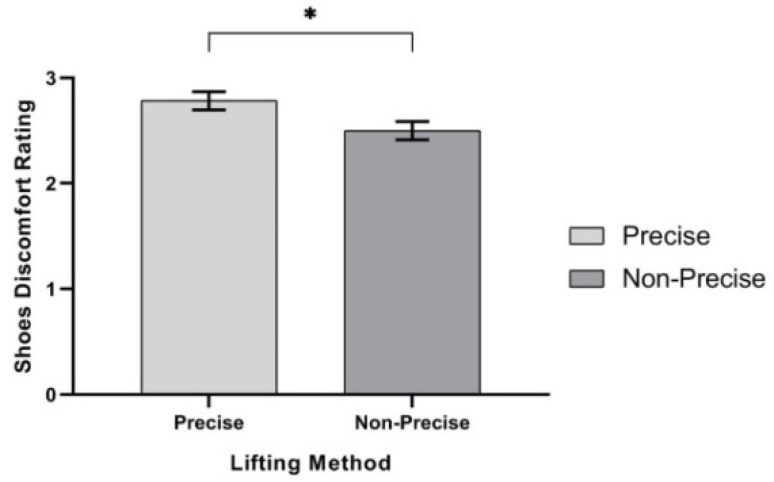
Effect of the lifting method on safety shoes discomfort rating. Bar errors represent the standard deviations. * *p* < 0.05.

**Figure 10 ijerph-17-03012-f010:**
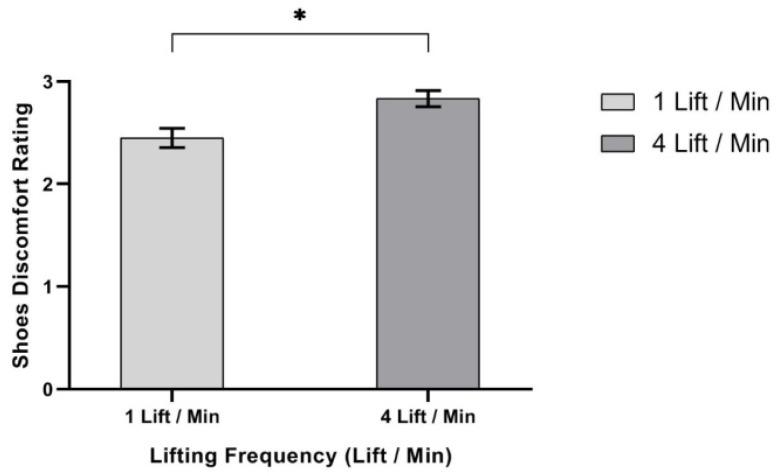
Effect of lifting frequency on safety shoes discomfort rating. Bar errors represent the standard deviations. * *p* < 0.05.

**Figure 11 ijerph-17-03012-f011:**
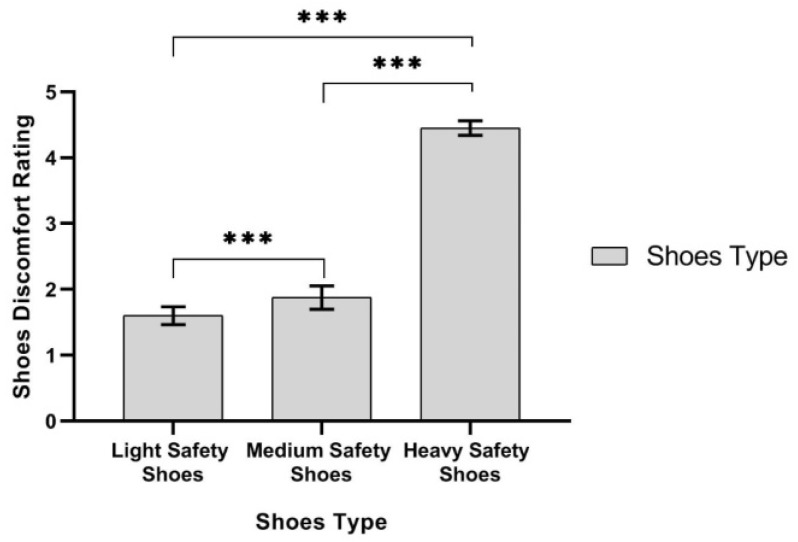
Effect of shoe type on safety shoes discomfort rating. Bar errors represent the standard deviations. *** *p* < 0.000.

**Table 1 ijerph-17-03012-t001:** Properties of the used safety shoes.

Measurement	Light Safety Shoes	Medium Safety Shoes	Heavy Safety Shoes
Upper	Full Leather with Double Density PU Sole	Genuine Full Leather with Double Density PU Sole	Waxy Full Grain Leather
Linings	Cambrele	Cambrele	Cambrele Woven
Tongue	…	Padded	Padded
Lacing	…	Through 4 Pairs Eyelets	Through 4 Pairs Eyelets
Collar	Padded	Padded	Padded
Toe Caps	Steel	Steel (Toe Cap)	Steel (Toe Cap)
Sole	Rubber	Polyurethane Molded	Polyurethane
Innersole	Full Sock	Full Sock	Full Sock
Cut	Low Cut	Low Cut	High Cut
Additional	…	…	Steel Plate
Weight	0.9 kg	1.05 kg	1.45 kg

**Table 2 ijerph-17-03012-t002:** The result of the analysis of variance (ANOVA) of the maximum acceptable weight of lift (MAWL).

Source	SS	DF	MS	F-Value	*p*-Value	Partial Eta Squared
LM	134.408	1	134.408	189.357	0.000	0.955
Error	6.388	9	0.710			
LF	222.496	1	222.496	284.832	0.000	0.969
Error	7.030	9	0.781			
ST	6.912	2	3.456	4.170	0.032	0.317
Error	14.918	18	0.829			

SS—sum square; DF—degree of freedom; MS—mean square; LM—lifting method; LF—lifting frequency; and ST—shoe type.

**Table 3 ijerph-17-03012-t003:** Summary of respirational rates (breaths/min) ANOVA result.

Source	SS	DF	MS	F-Value	*p*-Value	Partial Eta Squared
LF	649.605	1	649.605	43.530	0.000	0.829
Error	134.310	9	14.923			

**Table 4 ijerph-17-03012-t004:** Result of the ANOVA for minute ventilation (Ve(L/min)) response measured.

Source	SS	DF	MS	F-Value	*p*-Value	Partial Eta Squared
LF	853.653	1	853.653	67.861	0.000	0.883
Error	113.215	9	12.579			

**Table 5 ijerph-17-03012-t005:** Results of the ANOVA for V_CO2_ (mL/min) response.

Source (Variables)	SS	DF	MS	F-Value	*p*-Value	Partial Eta Squared
LF	1041731.907	1	1041731.907	60.375	0.000	0.870
Error	155289.517	9	17254.391			

**Table 6 ijerph-17-03012-t006:** Results of the ANOVA for V_O_2__/kg response.

Source (Variables)	SS	DF	MS	F-Value	*p*-Value	Partial Eta Squared
LF	250.765	1	250.765	52.163	0.000	0.853
Error	43.266	9	4.807			

**Table 7 ijerph-17-03012-t007:** Summary of heart rate (beats/min) ANOVA result.

Source	SS	DF	MS	F-Value	*p*-Value	Partial Eta Squared
LF	3514.622	1	3514.622	16.664	0.004	0.676
Error	1687.280	8	210.910			

**Table 8 ijerph-17-03012-t008:** The result of the ANOVA for safety shoes discomforts subjective response determined by the subject.

Source	SS	DF	MS	F-Value	*p*-Value	Partial Eta Squared
LM	2.408	1	2.408	5.898	0.038	0.396
Error	3.675	9	0.408			
LF	4.408	1	4.408	10.796	0.009	0.545
Error	3.675	9	0.408			
ST	197.717	2	98.858	100.063	0.000	0.917
Error	17.783	18	0.988			

**Table 9 ijerph-17-03012-t009:** Comparison among mean working heart rate (beats/min).

Source	Present Study	Ghaleb et al. [54]	Al Ashaik [38]	Singh et al. [45]	Ramadan [59]	Hafez [60]
1 lift/min	95.17	99.98	90.40	90.80	89.60	89.90
4 lifts/min	106.58	119.79	-	-	-	-
5 lifts/min	-	-	100.20	107.80	108.00	95.60

**Table 10 ijerph-17-03012-t010:** Effect of lifting frequency on the mean safety shoes discomfort rating (SSDR).

Lifting Frequency (lift/min)	Mean Safety Shoes Discomfort Rating (SSDR)	
	Present Study	Ghaleb et al. [54]
1 lift/min	2.83	3.67
4 lifts/min	2.50	3.88

**Table 11 ijerph-17-03012-t011:** Effect of safety shoe type on the mean safety shoes discomfort rating (SSDR).

Safety Shoe Type	Mean Safety Shoes Discomfort Rating (SSDR)		
	Present Study at Normal Lab Temperature	Ghaleb et al. [54]	Al Ashaik et al. [38] on 5 lifts/min at 20 °C
Light safety shoes	1.60	4.28	2.43
Medium safety shoes	1.88	3.93	2.43
Heavy safety shoes	4.50	3.13	3.79

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
