# Peer review of "Effects of Lifting Method, Safety Shoe Type, and Lifting Frequency on Maximum Acceptable Weight of Lift, Physiological Responses, and Safety Shoes Discomfort Rating"

_ijerph, 2020, doi:10.3390/ijerph17093012_

Round 1

Reviewer 1 Report

In order to improve the quality of the manuscript I make some comments to the authors.

Line 145- delete “standard deviation”, SD is not necessary to be expanded.

Study design is missed and also the date, place and deeper participants description.

Sample size calculation should be justified. It´s mandatory.

In addition, I miss the equator network guidelines.

Results and discussion section are ok.

Please, place the limitation section before the conclusion section.

The conclusions should be conclusive, please limit the section to the main findings.

Author Response

Response to the 1st  respected reviewer

Manuscript ID: ijerph-765461

Title: Effects of Lifting Method, Safety Shoe Type, and Lifting Frequency on Maximum Acceptable Weight of Lift, Physiological Responses, and Safety Shoes Discomfort Rating.

Greeting Goodness,

We post our thanks for your constructive comments and valuable suggestions concerning the quality of our manuscript. We have studied your comments and suggestions carefully and made our comments and response which we hope to meet with your expectation. Details of our comments and responses are listed in the following table for your review and approval. Moreover, the updated manuscript is resubmitted after correction showing all corrections in blue color.

Comments

Responses

Line 145- delete “standard deviation”, SD is not necessary to be expanded.

It is deleted.

Study design is missed and also the date, place and deeper participants description.

Study design is inserted in lines 145-154. Also, data and place are highlighted in blue lines in 165 to 159.

Sample size calculation should be justified. It´s mandatory.

How to calculate sample size is in lines 156-160.

In addition, I miss the equator network guidelines.

We did not understand what you mean by equator network guidelines.

Results and discussion section are ok.

Ok.

Please, place the limitation section before the conclusion section.

Done.

The conclusions should be conclusive, please limit the section to the main findings.

Done. Several sentences have been removed.

Reviewer 2 Report

Effects of Lifting Method, Safety Shoe Type, and Lifting Frequency on Maximum Acceptable Weight of Lift, Physiological Responses, and Safety Shoes Discomfort Rating

ijerph-765461

General Comments:

The manuscript presents data on subjective perception of maximum acceptable lifting weight, and subjective perception of discomfort rating of safety shoes worn as well as physiological responses that included VO2, VE, RR, CO2, HR.

The introduction provides sufficient information on the need for assessing the above mentioned dependent outcome variables, however there is no sufficient explanation for the independent variables, as to why the specific lifting frequencies were used.

The testing of only 10 participants is not statistically sound, especially with no explanation of a statistical sample size estimation. This has been addressed in the manuscript as a potential limitation.

The procedures are explained well with proper definitions of the variables tested and their corresponding analyses.  

The discussion of the results observed are appropriate.

The biggest concern I have is that, the manuscript very often, especially in the introduction and discussion/conclusion sections, say that there are no other studies that has looked at safety shoes, physiological responses and lifting frequencies. I think I have seen some work from the same lab and want to make sure that these statements are made appropriately.

Specific Comments:

Several sentence structure mistakes are present in the manuscript. Needs to full thorough review of these mistakes to address them. An example of the sentence structure mistakes that I point out is given below in line 482.

Line 482: Conclusion is misleading.

Lines 320-322: Suggest changing the color of the graph to grayscale similar to other figures in the manuscript.

All tables should have the abbreviations used explained under the table as footnotes.

Author Response

Response to the 2nd respected reviewer

Manuscript ID: ijerph-765461

Title: Effects of Lifting Method, Safety Shoe Type, and Lifting Frequency on Maximum Acceptable Weight of Lift, Physiological Responses, and Safety Shoes Discomfort Rating.

Greeting Goodness,

We post our thanks for your constructive comments and valuable suggestions concerning the quality of our manuscript. We have studied your comments and suggestions carefully and made our comments and response which we hope to meet with your expectation. Details of our comments and responses are listed in the following table for your review and approval. Moreover, the updated manuscript is resubmitted after correction showing all corrections in blue color.

Comments

Responses

The introduction provides sufficient information on the need for assessing the above mentioned dependent outcome variables; however there is no sufficient explanation for the independent variables, as to why the specific lifting frequencies were used.

Explained in details in lines 251-253.

The testing of only 10 participants is not statistically sound, especially with no explanation of statistical sample size estimation. This has been addressed in the manuscript as a potential limitation.

Done.

The procedures are explained well with proper definitions of the variables tested and their corresponding analyses.  

OK

The discussions of the results observed are appropriate.

Ok

The biggest concern I have is that, the manuscript very often, especially in the introduction and discussion/conclusion sections, say that there are no other studies that has looked at safety shoes, physiological responses and lifting frequencies. I think I have seen some work from the same lab and want to make sure that these statements are made appropriately.

Study of precise lifting capability while wearing safety shoe types is not seen in the literature. In addition, some references were added in line 148 and in references section.

Several sentence structure mistakes are present in the manuscript. Needs to full thorough review of these mistakes to address them. An example of the sentence structure mistakes that I point out is given below in line 482.

Considered. A native English speaking colleague is passed through the manuscript for English correction.

Line 482: Conclusion is misleading.

Done.

Lines 320-322: Suggest changing the color of the graph to grayscale similar to other figures in the manuscript.

Done.

All tables should have the abbreviations used explained under the table as footnotes.

Done.

Reviewer 3 Report

The reviewer understood that the present article mentioned findings on relationship between Maximum Acceptable Weight of Lift and safety shoe conditions. Research questions on safety shoes conditions (weight and discomfort ratings) are quite unique and results are valuable for public health. The paper is well organized, and results are discussed based on objective data. The reviewer have some comments to improve this paper as follows:

(1) Line 94
It is the first time to use the abbreviation, MAWL. At the first, please describe as "Maximum Available Weight of Lift (MAWL)."

(2) Line 150
It is the first time to use the abbreviation, SAR. Please describe what "SAR" stands for.

(3) 2.6 Experimental Procedures
Please describe the way to select the shoe size for the participant's foot length.

(4) Line 400
TYPO sheo -> shoe

Author Response

Response to the 3rd   respected reviewer

Manuscript ID: ijerph-765461

Title: Effects of Lifting Method, Safety Shoe Type, and Lifting Frequency on Maximum Acceptable Weight of Lift, Physiological Responses, and Safety Shoes Discomfort Rating.

Greeting Goodness,

We post our thanks for your constructive comments and valuable suggestions concerning the quality of our manuscript. We have studied your comments and suggestions carefully and made our comments and response which we hope to meet with your expectation. Details of our comments and responses are listed in the following table for your review and approval. Moreover, the updated manuscript is resubmitted after correction showing all corrections in blue color.

Comments

Responses

(1) Line 94        It is the first time to use the abbreviation, MAWL. At the first, please describe as "Maximum Available Weight of Lift (MAWL)."

Done.

(2) Line 150       It is the first time to use the abbreviation, SAR. Please describe what "SAR" stands for.

Described.

(3) 2.6 Experimental Procedures   Please describe the way to select the shoe size for the participant's foot length.

It is explained in lines 264-265.

(4) Line 400    TYPO sheo -> shoe

Corrected.

Round 2

Reviewer 1 Report

I think all the authors have been implemented all the recommendations and suggestions.

Thus, I think that this manuscript is suitable to be published in the present form.

Author Response

Response to the 1st  respected reviewer

Manuscript ID: ijerph-765461

Title: Effects of Lifting Method, Safety Shoe Type, and Lifting Frequency on Maximum Acceptable Weight of Lift, Physiological Responses, and Safety Shoes Discomfort Rating.

Greeting Goodness,

We post our thanks again for your constructive comments and valuable suggestions concerning the quality of our manuscript. We have studied your comments and suggestions carefully and made our comments and response which we hope to meet with your expectation. Details of our comments and responses are listed in the following table for your review and approval. Moreover, the updated manuscript is resubmitted after correction showing all corrections in blue color.

Comments

Responses

I think all the authors have been implemented all the recommendations and suggestions.

Thank you very much for good gesture.

Thus, I think that this manuscript is suitable to be published in the present form.

Thank you very much.

Reviewer 2 Report

Authors have addressed comments. I do suggest refining some discussion points and elaborating on the limitations. The authors state that lifting frequency and safety shoes are not available in the literature. But please make sure that is specified clearly. Is there any lifting frequency studies independent of the safety shoes? If so, they need to be referenced in the manuscript. 

Author Response

Response to the 2nd respected reviewer

Manuscript ID: ijerph-765461

Title: Effects of Lifting Method, Safety Shoe Type, and Lifting Frequency on Maximum Acceptable Weight of Lift, Physiological Responses, and Safety Shoes Discomfort Rating.

Greeting Goodness,

We post our thanks again for your constructive comments and valuable suggestions concerning the quality of our manuscript. We have studied your comments and suggestions carefully and made our comments and response which we hope to meet with your expectation. Details of our comments and responses are listed in the following table for your review and approval. Moreover, the updated manuscript is resubmitted after correction showing all corrections in blue color.

Comments

Responses

Authors have addressed comments. I do suggest refining some discussion points and elaborating on the limitations.

We did carefully reviewed the discussion section, we found every sentence is necessary to be include, otherwise we will lost important information, except one sentence it has been removed. It was in line 456. In addition, a sentence has been added in the limitation in lines 500-503.

The authors state that lifting frequency and safety shoes are not available in the literature. But please make sure that is specified clearly.

It is cleared in lines 148, and 149.

Is there any lifting frequency studies independent of the safety shoes? If so, they need to be referenced in the manuscript.

It has been indicated in line 147.

Reviewer 3 Report

Authors revised adequately about all reviewers' comments. I think that the present paper can be accepted to the journal. It contains valuable knowledge in this academic field.

Author Response

Response to the 3rd   respected reviewer

Manuscript ID: ijerph-765461

Title: Effects of Lifting Method, Safety Shoe Type, and Lifting Frequency on Maximum Acceptable Weight of Lift, Physiological Responses, and Safety Shoes Discomfort Rating.

Greeting Goodness,

We post our thanks again for your constructive comments and valuable suggestions concerning the quality of our manuscript. We have studied your comments and suggestions carefully and made our comments and response which we hope to meet with your expectation. Details of our comments and responses are listed in the following table for your review and approval. Moreover, the updated manuscript is resubmitted after correction showing all corrections in blue color.

Comments

Responses

Authors revised adequately about all reviewers' comments. I think that the present paper can be accepted to the journal. It contains valuable knowledge in this academic field.

Thank you very much for the acceptance.